# Highly Porous Carbon Aerogels for High-Performance Supercapacitor Electrodes

**DOI:** 10.3390/nano13050817

**Published:** 2023-02-23

**Authors:** Jong-Hoon Lee, Seul-Yi Lee, Soo-Jin Park

**Affiliations:** Department of Chemistry, Inha University, Incheon 22212, Republic of Korea

**Keywords:** carbon aerogel, supercapacitor, sol–gel polymerization, electric double-layer capacitors, physical activation

## Abstract

In recent years, porous carbon materials with high specific surface area and porosity have been developed to meet the commercial demands of supercapacitor applications. Carbon aerogels (CAs) with three-dimensional porous networks are promising materials for electrochemical energy storage applications. Physical activation using gaseous reagents provides controllable and eco-friendly processes due to homogeneous gas phase reaction and removal of unnecessary residue, whereas chemical activation produced wastes. In this work, we have prepared porous CAs activated by gaseous carbon dioxide, with efficient collisions between the carbon surface and the activating agent. Prepared CAs display botryoidal shapes resulting from aggregation of spherical carbon particles, whereas activated CAs (ACAs) display hollow space and irregular particles from activation reactions. ACAs have high specific surface areas (2503 m^2^ g^−1^) and large total pore volumes (1.604 cm^3^ g^−1^), which are key factors for achieving a high electrical double-layer capacitance. The present ACAs achieved a specific gravimetric capacitance of up to 89.1 F g^−1^ at a current density of 1 A g^−1^, along with a high capacitance retention of 93.2% after 3000 cycles.

## 1. Introduction

With the continuously increasing industrial development, energy shortage and global warming have become serious issues; in this context, the identification of clean energy sources and efficient energy storage technologies for low-carbon and sustainable economic development has become an important task [1,2,3]. A supercapacitor is a more effective type of energy storage device compared with traditional physical capacitors and batteries [4,5]. In comparison with conventional capacitors, supercapacitors use materials with a higher specific surface area as electrodes, as well as thinner dielectrics, resulting in a greater specific capacitance and energy density. Furthermore, compared with batteries, the charge storage mechanism of supercapacitors is based on the surface reactions of the electrode material, and no diffusion of ions takes place inside the material; therefore, supercapacitors provide higher power densities within the same volume [6,7]. Supercapacitors are generally classified into two categories depending on the energy storage mechanisms as electric double-layer capacitors (EDLCs) and pseudocapacitors [8,9,10,11]. EDLCs are based on the confrontation of charges at the electrode/solution interface by the alignment of electrons or ions. For an electrode/solution system, a double layer is formed at the interface between the electrically conductive electrodes and the ion-conductive electrolyte solution. When an electric field is applied to the two electrodes, the anions in the solution migrate to the positive and negative electrodes, forming a double layer on their surfaces [12]. After removing the electric field, the positive and negative charges on the electrodes attract the oppositely charged ions in the solution, which produce a relatively stable potential difference between the positive and negative electrodes [13]. For each electrode, a specific amount of heterogeneous ionic charge equal to the charge on the electrode will be generated within a certain distance (dispersion layer) to keep the electrical neutrality. When the two electrodes are connected to an external circuit, the charges on the electrodes migrate to generate a current in the external circuit, and the ions in the solution migrate to the electrode surface, which becomes electrically neutral; therefore, enhanced accessibility to large electrolyte ions associated with a high specific surface area is required to achieve high-performance EDLCs [14,15,16].

Therefore, many researchers have focused on the adsorption process using various porous materials, such as activated carbons, zeolites, metal–organic frameworks (MOFs), and porous polymers [17,18,19,20,21]. Carbonaceous materials have many advantages, such as a relatively low cost, thermal/chemical stability, and tunable pore characteristics to meet application-specific requirements [22,23,24]. In particular, carbon aerogels (CAs) are synthetic porous gel-type materials, in which voids occupy over 90% of the entire volume. This structure remains a three-dimensional and highly porous skeleton without structural collapse [25]. The highly porosity of CAs endows them with diverse unique characteristics, such as ultralow density, low thermal conductivity, and large specific surface area. These remarkable properties provide various applications of CAs, such as adsorbents, catalytic supports, and energy storage materials. The highly porous structure of CAs and their broad pore size distribution, ranging from micropores to mesopores, are suitable for electronic and energy storage applications. Micropores (<2 nm) generally provide abundant adsorption sites for ions providing high specific capacitance of electrodes, whereas mesopores (2–50 nm) enable the rapid diffusion of electrolyte ions which correlate with rate capability and cycling stability of electrodes [26,27]. Fierro et al. reported distribution of mesopores to fast ion diffusion. From comparison with mesopore-dominant and micropore-dominant species, mesoporous carbons represent 75% rate capability whereas microporous carbons show 50% rate capability. This result represents wide porous structure providing easy ion mobility and lower diffusion resistance [28]; therefore, well-developed CAs with both micropores and mesopores provide high specific surface areas, leading to high ion adsorption capacities and easily accessible pathways for electrolyte ions [29].

To improve the porosity of carbonaceous materials, many studies have recently attempted to increase the specific surface area and pore volume of the adsorbents via activation processes [30,31,32,33]. Two methods have been suggested for this purpose: chemical and physical activation, both of which develop the porosity of carbon materials by forming micropores and narrow-size mesopores. In chemical activation approaches, carbonaceous materials are mixed with chemical agents, such as alkali salts (sodium hydroxide (NaOH), potassium hydroxide (KOH), and zinc chloride (ZnCl_2_)) [34,35,36,37,38,39]. Chemical activation exhibits excellent performance in terms of porosity development, owing to the deep penetration of the activating reagents into the carbon lattice during the activation process [40]. Nevertheless, chemical activation not only causes environmental pollution and incurs high costs, associated with the removal of the remaining residue after activation, but also renders it difficult to control the degree of activation. On the other hand, in physical activation, suitable oxidizing gas-phase agents (oxygen (O_2_), carbon dioxide (CO_2_), and steam (H_2_O)) are employed to produce porosity, typically at a high temperature [41,42]. Physical activation is environmentally friendly and renders it easy to control the degree of activation, because the residue removal process is not needed and the activating reagent is a homogeneous gas phase.

On the other hand, supercapacitors are classified according to the structure such as 1D yarn-, 2D sheet-, and 3D thick structured types. A thick electrodes supercapacitor provides higher energy density compared with a thin film type supercapacitor, because bulk species can increase the loading of the active materials providing higher ion storage sites for charge; however, the augmentation of electrode thickness may pose challenges in terms of transporting electrons and ions in the direction of the electrode thickness and separator. Consequently, practical applications of thick electrode devices are confronted with difficulties and complexities in both transmission and manufacturing. In this regard, introducing large pore for fast ion diffusion is one of strategy to overcome the challenges of thick electrodes [43,44,45].

In this work, activated CA (ACA) supercapacitor electrodes were fabricated by polymerization and physical activation processes. CAs were physically activated by pressure-assisted CO_2_ to produce well-developed pore structures and, ultimately, increase the electrochemical capacitances. The ACA electrodes exhibited higher specific capacitances compared with CA electrodes, owing to their higher porosity that facilitated ion accumulation and electrostatic adsorption of electrolytic ions. Furthermore, the mesopores formed by pore expansion upon a further activation process provided efficient electron pathways, resulting in a higher rate capability and cycling stability. ACA activated under 60 min exhibited the highest specific capacitance, a rate capability under 1–5 A g^−1^, and cycling stability after 3000 cycles of 89.1 F g^−1^, 82.6%, and 93.2%, respectively.

## 2. Materials and Methods

### 2.1. Materials

Resorcinol (7.5 g, >99%, Sigma-Aldrich, St. Louis, MO, USA), formaldehyde solution (37 wt.% in H_2_O with 10% methanol stabilizer, Sigma-Aldrich, St. Louis, MO, USA), acetone (>99.8%, Daejung Chemicals, Siheung-si, Republic of Korea), sodium carbonate (>99.5%, Sigma-Aldrich, St. Louis, MO, USA), carbon black (Ketjenblack, EC-600JD, AkzoNobel, Amsterdam, The Netherlands), polyvinylidene fluoride (PVDF, *M*_W_~534,000, Sigma-Aldrich, St. Louis, MO, USA), sulfuric acid (>95%, H_2_SO_4,_ Sigma-Aldrich, St. Louis, MO, USA), and 1-methyl-2-pyrrolidone (NMP, >99%, Sigma-Aldrich, St. Louis, MO, USA).

### 2.2. Synthesis of Carbon Aerogels

The CAs were prepared by sol–gel polymerization of resorcinol and formaldehyde. Resorcinol (7.5 g) was dissolved in distilled water (75 mL) and formaldehyde solution (25 g). Sodium carbonate, used as a base catalyst, was subsequently added to the solution and stirred for 3 h. The obtained mixture was dried for 3 h at room temperature. After that, the wet gel was dried at 80 °C for 48 h and then immersed in acetone solution for 24 h without stirring. The aerogels were obtained after freeze-drying and carbonized in a furnace with a heating rate of 2 °C min^−1^ under N_2_ gas atmosphere. The obtained CAs were subsequently activated at 900 °C in a furnace under CO_2_ gas flow (heating rate 5 °C min^−1^, flow rate 100 mL min^−1^) for 30, 45, 60, and 75 min, with the corresponding products denoted as ACA30, ACA45, ACA60, and ACA75, respectively. The obtained ACAs were washed with distilled water until neutral pH and dried in an oven at 80 °C for 24 h.

### 2.3. Characterizations

Surface morphology images were obtained by scanning electron microscope (SEM) using an SU-8010 instrument (Hitachi Co., Tokyo, Japan). To investigate the porous properties of the samples, N_2_/77 K adsorption–desorption isotherms were measured using a BELSORP–Max instrument (BEL Japan, Inc., Tokyo, Japan). The samples were degassed at 473 K for 6 h to reduce the residual pressure below 10^−6^ bar. Specific surface areas (*S*_BET_) were calculated using the Brunauer–Emmett–Teller (BET) equation. Pore size distributions were obtained using non-local density functional theory (NLDFT) for modeling the N_2_/77 K adsorption with a slit pore kernel. X-ray photoelectron spectroscopy (XPS) and Raman spectroscopy were investigated by Kα (Thermo Fisher Scientific Inc., Waltham, NJ, USA) with a monochromatic Al Kα X-ray source and LabRAm HR Evolution (Horiba, Ltd., Tokyo, Japan), respectively.

### 2.4. Electrochemical Performance of Samples

The electrochemical performances of the prepared samples were characterized by cyclic voltammetry (CV) and galvanostatic charge/discharge (GCD) measurements using an IviumStat electrochemical workstation (Ivium Technologies, The Netherlands). All electrochemical properties were investigated using 1 M H_2_SO_4_ as electrolyte in a three-electrode system, with a platinum coil as the counter electrode and Ag/AgCl as the reference electrode. The working electrode consisted of 80 wt.% active material (ACA), 10 wt.% conductive material (carbon black), 10 wt.% binder (PVDF), and solvent (NMP). After stirring for 12 h, the product of 0.4 g was pressed onto a nickel foil of 1 cm×4 cm size. The prepared electrodes were dried at 60 °C in air for 12 h to remove the NMP solvent. The CV curves of the working electrode were obtained at scan rates of 10, 20, 30, 50, and 100 mV s^−1^ in the voltage range from −0.2 to 0.8 V. GCD curves were obtained from −0.2 to 0.8 V at current densities of 1, 2, 3, 4, and 5 A g^−1^.

## 3. Results and Discussion

Figure 1 shows a schematic diagram of the fabrication of the ACAs. In the gelation step during the carbon aerogel synthesis, a hydrogel was formed by polymerization and crosslinking between resorcinol and formaldehyde, with sodium carbonate as base catalyst. The resorcinol anions, which are capable of strong nucleophilic addition, are formed during the gelation process under basic conditions [46]; therefore, resorcinol was chemically crosslinked with formaldehyde in the presence of the base catalyst [47]. The drying process significantly influenced the final carbon aerogel products. The highly porous structure of the aerogels mostly collapsed during the drying process, due to surface tension and capillary forces [48]. Freeze-drying is an efficient method to minimize these destructive forces during solvent evaporation, in which the solvent is removed via sublimation at low pressure, without the formation of a gas–liquid interface. The mechanically stabilized aerogel obtained after the freeze-drying process was subsequently carbonized under an inert N_2_ atmosphere. To maximize the porosity and introduce micropores, the obtained CAs were physically activated using CO_2_ as activating reagents.

Figure 2 shows SEM images of pristine CA and ACA samples. A botryoidal shape was observed for pristine CA, owing to the aggregation of spherical carbon particles via polymerization and crosslinking. The surface morphology changed after the CO_2_ activation process, with the appearance of hollow spaces and irregular particles due to the reaction of carbon with CO_2_ during the activation process; moreover, we observed a decrease in the number of relatively large interparticle voids, together with a preference for the formation of smaller pores.

Figure 3 displays N_2_ adsorption–desorption isotherms and pore size distributions. The increased initial adsorption with increasing activation time reflected the higher number of micropores resulting from the activation process. The ACA samples exhibited type IV isotherms according to the IUPAC classification, with typical hysteresis loops at relative pressures above 0.4 [49]. These features were attributed to the capillary condensation of the nitrogen molecules within the mesopores. The adsorbed volume increased with increasing relative pressure, because of the monolayer/multilayer adsorption of the nitrogen molecules on both the mesopores and macropores. The adsorption capacity of ACA60 increased rapidly at relative pressures above 0.4, because of its large mesopore content. After an activation time of 60 min, as in the case of ACA75, the adsorbed N_2_ amount decreased compared with that of ACA60, due to the collapse of the porous structure upon excessive activation. Based on the pore size distribution calculated by the NLDFT model, pore size expansion occurred with increasing activation time. The pore size distribution of ACA75 showed a decreased number of larger pores compared with ACA60, reflecting the collapse of large pores under excessive activation conditions.

Table 1 lists the textural properties of the prepared samples, as obtained from the N_2_ adsorption–desorption isotherms. The micropore volumes of the prepared samples were estimated by the Dubinin–Radushkevich (D–R) model (Equation (1)) [50]:(1)$$W=W_{0}exp\left[-B{\left(\frac{T}{\beta }\right)}^{2}log^{2}\left(\frac{P_{0}}{P}\right)\right]$$
where *W* indicates the volume of the liquid adsorbed at *P P*_0_^–1^, *W*_0_ is the total volume of the adsorbates in the micropores, while *B* and *β* are the adsorbent and adsorbate constants, respectively.

The as-received ACA had a high specific surface area (2503 m^2^ g^−1^) and total pore volume (1.604 cm^3^ g^−1^). After activation, the specific surface area and micropore volume generally increased with increasing the activation time, up to 60 min (ACA60). These results indicate that the CO_2_ activation of ACA led to the development of microporosity. Activation times over 60 min resulted in a decrease of the specific surface area and total pore volume, due to the excessive activation leading to the collapse of the porous structure (ACA75).

The fraction of micropore volume of all samples gradually decreased until the activation time reached 60 min. This is because, as shown in Equations (2)–(6), CO_2_ gas molecules intercalated into the carbon lattice, causing the expansion of the existing micropores into mesopores during activation [51]; thus, ACA60 had the highest specific surface area (2503 m^2^ g^−1^) and micropore volume (1.604 cm^3^ g^−1^). On the other hand, ACA75 showed a decreased micropore volume and consisted almost entirely of micropores, which led to a higher micropore volume fraction, owing to the excessive activation with CO_2_.
(2)$$C+{\mathrm{CO}}_{2}\to 2\mathrm{CO}\;\;\;\Delta H=+173\;{\mathrm{kJ}}^{-1}\mathrm{mol}$$
(3)$$C^{*}+{\mathrm{CO}}_{2}\to C^{*}\left({\mathrm{CO}}_{2}\right)$$
(4)$$C^{*}+\left({\mathrm{CO}}_{2}\right)⇌C^{*}\left(O\right)+\mathrm{CO}$$
(5)$$C^{*}\left(O\right)⇌\mathrm{CO}$$
(6)$$\mathrm{CO}+C^{*}\left(O\right)⇌C^{*}\left({\mathrm{CO}}_{2}\right)$$

The chemical properties of samples are shown in Figure 4. The elemental compositions by XPS measurement represent increasing oxygen component as 0.05 to 0.21 at.%, which represent oxygen functionalization via activation reaction progress. Raman spectroscopy exhibited two characteristic peaks of 1340 and 1579 cm^−1^ attributed to the D-band and G-band, respectively. The G-band indicates in-plane (E_2g_) stretching of sp^2^ graphitic structure, whereas D-band is associated with the defect of graphitic structure. The intensity ratio of D- and G-band (I_D_/I_G_) are calculated to evaluate the degree of defects in carbon surfaces. The ID/IG ratios for CA, ACA30, ACA45, ACA60, and ACA75 are 0.89, 1.00, 1.08, 1.12, and 1.20, respectively. The calculated values represent high defects attributed to progression of activation reaction.

The electrochemical properties of the ACA series for electrochemical capacitor applications were investigated using a three-electrode system. Figure 5a shows the CV curves of the as-prepared samples, obtained in the potential window between −0.2 and 0.8 V at a scan rate of 10 mV s^−1^. The CV curves of the ACAs presented a nearly regular rectangular shape without obvious redox peaks, representing an electrical double-layer capacitive behavior. The imperfect rectangular shapes of CV curves result from oxygen moieties by activation process which was identified in Figure 4. The areas of the CV curves of the ACAs were larger than those of the pristine CAs, indicating that the specific capacitance of the ACAs was substantially increased compared with that of the less porous CAs. Figure 5b shows the CV curves of the as-prepared samples at different voltage scan rates. The CV curve collected at a high scan rate retained its rectangular shape compared to those obtained at lower scan rates, indicating the sustained reversible adsorption of electrolyte ions at the electrode–electrolyte interface.

Figure 6a shows the GCD curves of the ACA series at a current density of 1 A g^−1^. The curves exhibit symmetric triangular shapes, which indicate an ideal supercapacitor behavior attributed to non-faradaic reactions. The GCD curves measured for the ACA60 electrode showed a decreasing trend with increasing discharge current density because it was hard for the electrolyte ions to penetrate the interior of the active material at high current density, owing to the slow ion diffusion kinetics (Figure 6b). The specific capacitance of the as-prepared samples was calculated according to Equation (7) [52,53]:(7)$$C_{m}=\frac{I\times t}{\Delta V\times m}$$
where *C_m_* is the specific capacitance (F g^−1^), *I* is the charge/discharge current, *t* is the charge/discharge time, ΔV is the potential window, and *m* is the mass of active material. The highest specific capacitance, obtained for ACA60, was 89.1 F g^−1^ at 1 A g^−1^. This excellent performance resulted from the appropriate combination of ACA60. Figure 6c represents the Nyquist plot of CA and ACA60 for electrochemical impedance analysis. An ideal supercapacitor exhibits a small series resistance and a charge-transfer resistance, representing a high-frequency semicircle followed by a sloped line at a low frequency. The behavior of ion diffusion is observed in the form of a straight line in the low frequency region, and a more vertical line indicates better double-layer behavior. A larger semi-arc radius indicates a greater charge-transfer resistance, while a smaller radius suggests a lower charge-transfer resistance. Moreover, the intrinsic resistance of the electrode can be estimated by the x-intercept of the curve on the high-frequency region representing the smaller the x-intercept at the smaller intrinsic resistance. This can be attributed to the presence of Warburg impedance and diffusive resistance, which is primarily caused by ion transport onto the electrode surface. In addition, the visible high-frequency semicircle observed for all samples is likely a result of charge-transfer resistance at the electrode surface [54,55]. ACA60 represents smaller intercept indicating low intrinsic resistance compared with CA. This was attributed to the significant improvement in highly porous structure for fast ion diffusion and hydrophilicity of CA after activation, which led to a smaller charge-transfer resistance in the ACA electrode. In addition, the ACA60 exhibited more vertical straight line in the low frequency region compared with CA representing better double-layer behavior.

The specific capacitances obtained in this work and in other studies reported in the literature are compared in Table 2 and Table 3 [56,57,58,59]. Commercial porous carbon-based (YP-50F, Kuraray) represented specific capacitance of 64 F g^−1^ using 1M TEABF_4_/PC electrolyte at potential window of 0–2.5 V and scan rate of 1 A g^−1^. Carbon aerogels of other reports represented specific capacitance of 43–108 F g^−1^. Even though large potential window and using organic and high concentration electrolyte, H_3_PO_4_-modified carbon aerogels and cellulose fibrous aerogels achieved specific capacitance of 103 and 108 F g^−1^, respectively, which differed little from this work.

Figure 6b shows the GCD curve of the ACA electrode at various current densities (1 to 5 A g^−1^), showing that the specific capacitance of the electrode decreased with increasing discharge current density. This is mainly because of the electrolyte ions diffuse slowly at high current densities and cannot penetrate the active material to a sufficient extent.

The rate capability of the prepared electrodes was determined by measuring the capacitance at different current densities (Figure 7a). The specific capacitance retention was evaluated at applied current densities increasing from 1 to 5 A g^−1^. The specific capacitances of the CA, ACA30, ACA45, ACA60, and ACA75 samples decreased from 21.2, 56.7, 59.6, 89.1, and 62.3 to 14.8, 43.8, 45.3, 73.6, and 49.1 F g^–1^, indicating retentions of 69.8, 75.5, 76.0, 82.6, and 78.8%, respectively. The higher rate capability of ACA60 was attributed to the facile ion diffusion pathway associated with the large mesopores. The cycling stability of electrode materials is important for their practical application in energy storage devices. The cycling performance of electrodes consisting of ACA60 was investigated by charging and discharging the capacitor 3000 times at a current density of 3 A g^−1^. ACA60 retained 93.2% of its initial specific capacitance value (Figure 7b).

## 4. Conclusions

In this work, carbon aerogel-based porous supercapacitors electrodes were fabricated by polymerization and physical activation processes. The ACA electrodes exhibited higher specific capacitances, owing to the micro- and meso-porous 3D structure of ACA, which provided a higher porosity facilitating the ion accumulation and electrostatic adsorption of electrolytic ions; furthermore, the mesopores formed by pore expansion upon a further activation process provided a facile electron pathway, resulting in a higher rate capability and cycling stability. The ACA electrodes achieved a specific capacitance, rate capability under 1–5 A g^−1^, and cycling stability after 3000 cycles of 89.1 F g^−1^, 82.6%, and 93.2%, respectively.

## Figures and Tables

**Figure 1 nanomaterials-13-00817-f001:**
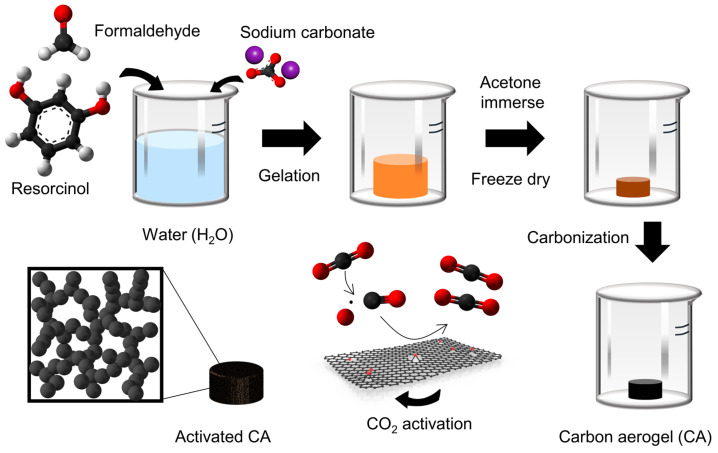
Schematic illustration of the preparation of activated carbon aerogel (ACA) samples.

**Figure 2 nanomaterials-13-00817-f002:**
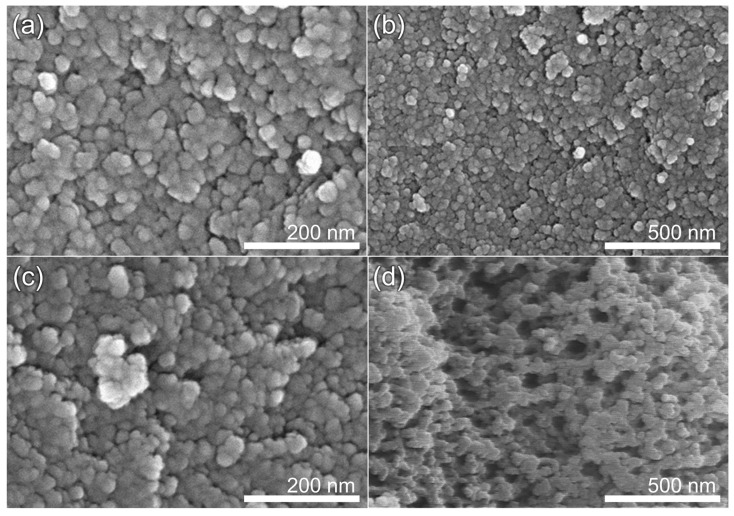
SEM images of CA and ACA samples: (**a**,**b**) high- and low-magnification images of CA and (**c**,**d**) high- and low-magnification images of ACA60.

**Figure 3 nanomaterials-13-00817-f003:**
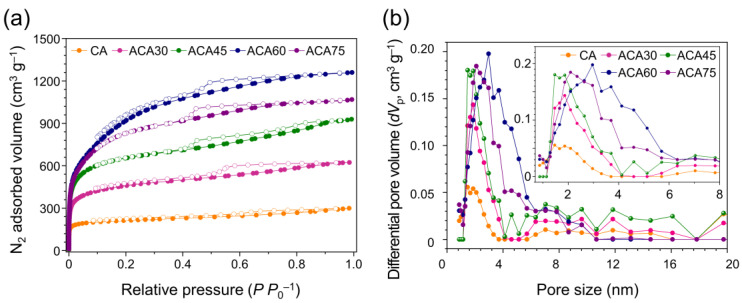
(**a**) N_2_ adsorption–desorption isotherms at 77 K (filled circles: adsorption and open circles: desorption) and (**b**) pore size distributions of prepared samples (inset: magnification between pore size of 0.7 and 8 nm).

**Figure 4 nanomaterials-13-00817-f004:**
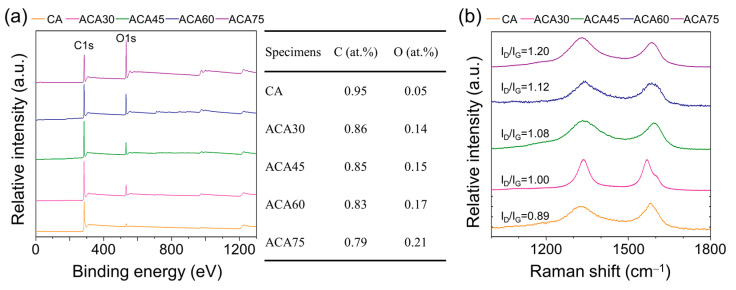
(**a**) XPS graphs with elemental composition (evaluated by XPS measurement) of samples and (**b**) Raman spectrum of samples with I_D_/I_G_ ratio.

**Figure 5 nanomaterials-13-00817-f005:**
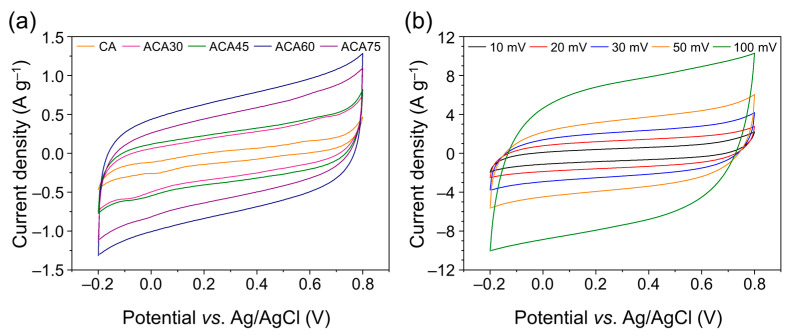
(**a**) CV curves of CA and ACA electrodes collected at a scan rate of 10 mV s^−1^ and (**b**) CV curves of ACA60 electrode at different scan rates.

**Figure 6 nanomaterials-13-00817-f006:**
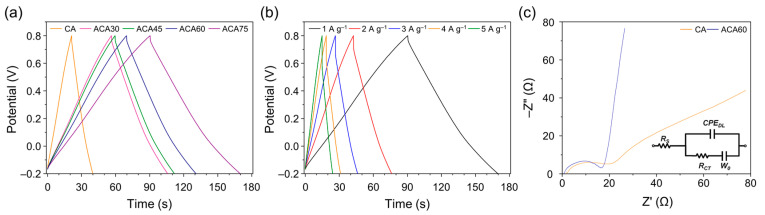
(**a**) GCD curves of CA and ACA electrodes obtained at a current density of 1 A g^−1^, (**b**) GCD curves of ACA60 electrode obtained at different current densities (1–5 A g^−1^), and (**c**) Nyquist plots of CA and ACA60.

**Figure 7 nanomaterials-13-00817-f007:**
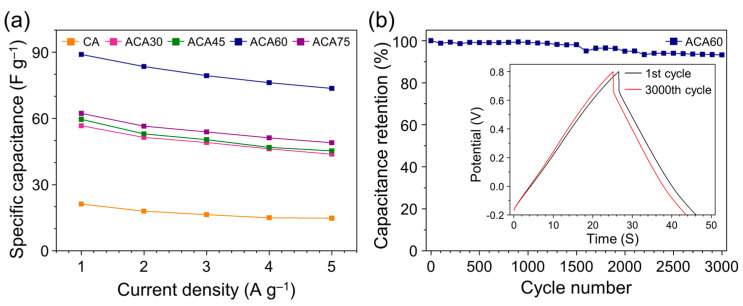
(**a**) Comparison of specific capacitances of CA and ACA samples and (**b**) cycle performance of ACA60 electrode at a current density of 3 A g^−1^ over 3000 cycles (inset: GCD curves for 1st and 3000th cycle).

**Table 1 nanomaterials-13-00817-t001:** Textural properties of prepared samples.

Specimen	*S*_BET_ ^a^ (m^2^ g^–1^)	*V*_total_ ^b^ (cm^3^ g^–1^)	*V*_micro_ ^c^ (cm^3^ g^–1^)	*V*_meso_ ^d^ (cm^3^ g^–1^)	*F*_micro_ ^e^ (%)
CA	704	0.381	0.306	0.075	80.3
ACA30	1436	0.794	0.596	0.198	75.1
ACA45	2094	1.182	0.862	0.320	72.9
ACA60	2503	1.604	1.132	0.472	70.6
ACA75	2403	1.359	0.968	0.391	71.2

^a^ *S*_BET_: specific surface area calculated using BET equation in a relative pressure range of 1·10^–4^–0.05. ^b^ *V*_total_: total pore volume estimated at a relative pressure *P P*_0_^–1^ = 0.99. ^c^ *V*_micro_: micropore volume determined from the D–R equation. ^d^ *V*_meso_: mesopore volume determined by subtracting *V*_micro_ from the total pore volume. ^e^ *F*_micro_: fraction of micropore volume, calculated as (micropore volume/total pore volume) × 100.

**Table 2 nanomaterials-13-00817-t002:** Specific capacitances of samples (F g^−1^).

Specimen	1 A g^−1^	2 A g^−1^	3 A g^−1^	4 A g^−1^	5 A g^−1^
CA	21.2	18.0	16.4	15.0	14.8
ACA30	56.7	51.4	49.1	46.2	42.8
ACA45	59.6	53.0	50.4	46.9	45.3
ACA60	89.1	83.5	79.4	76.3	73.6
ACA75	62.3	56.5	53.9	51.2	49.1

**Table 3 nanomaterials-13-00817-t003:** Comparison of the specific capacitance with other literatures.

Materials ^a^	Electrolyte ^b^	Potential Window (V)	Current Density/Scan Rate	Specific Capacitance (F g^−1^)	References
YP-50F	1M TEABF_4_/PC	0–2.5	1 A g**^−1^**	64	[56]
H_3_PO_4_-modified carbon aerogels	1M TEABF_4_/PC	0–2.5	1 A g**^−1^**	108	[56]
Cellulose fibrous aerogels	6M KOH	0–1	10 mV s**^−1^**	103	[57]
Steam-activated hard carbons	6M KOH	−0.3–0.7	1 A g**^−1^**	68	[58]
WS_2_ (TMD)-loaded carbon aerogels	1M Na_2_SO_4_	0.1–0.9	1 A g**^−1^**	43	[59]
Steam-activated carbon aerogel (ACA60)	1M H_2_SO_4_	−0.2–0.8	1 A g**^−1^**	89	This work

^a^ TMD: transition metal dichalcogenide. ^b^ TEABF_4_: tetra ethyl ammonium tetrafluoroborate, PC: polycarbonate.

## Data Availability

The data presented in this study are available on request from the corresponding author.

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
