# Peer review of "Highly Porous Carbon Aerogels for High-Performance Supercapacitor Electrodes"

_nanomaterials, 2023, doi:10.3390/nano13050817_

Round 1
Reviewer 1 Report
n this manuscript, authors prepared carbon aerogels as thick electrode for supercapacitors. After activation, the carbon aerogels showed high specific surface areas and large total pore volume. The resultant carbon aerogel electrode also possessed good supercapacitance performance. In general, it is an interesting work. However, there are still some issues to be addressed. A moderate revision is necessary before its acceptance.
1. The carbon aerogels as electrode can be classified into the thick electrode. More introduction on this point should be included in the introduction section.
2. When generally introduce the supercapacitor, more recent and important articles should be included to support the statements: Nanocellulose and its derived composite electrodes toward supercapacitors: Fabrication, properties, and challenges; Recent progress in carbon-based materials for supercapacitor electrodes: a review; Chitin derived nitrogen-doped porous carbons with ultrahigh specific surface area and tailored hierarchical porosity for high performance supercapacitors; Design and fabrication of conductive polymer hydrogels and their applications in flexible supercapacitors; etc.
3. In the last paragraph of introduction, please carefully recheck the data.
4. One more sub-section on materials should be added.
5. How about the size of the carbon aerogels for electrode? More details on the preparation of electrode for characterizations should be provided.
6. More activation approaches including alkali salts and zinc chloride should be further clarified more supporting relevant articles: ZnCl 2 regulated flax-based porous carbon fibers for supercapacitors with good cycling stability; Pyrolysis of zinc salt-treated flax fiber: Hierarchically porous carbon electrode for supercapacitor; One step activation by ammonium chloride toward N-doped porous carbon from camellia oleifera for supercapacitor with high specific capacitance and rate capability; etc.
7. TEM images are suggested to show the smaller pore structures.
8. How about the SEM images with smaller magnifications to show the larger pore structures?
9. Three-line table should be used for a more scientific expression.
10. More comparison on the supercapacitance is necessary with supporting articles: Chinese Chemical Letters 31 (7), 1986-1990, 2020; Polymer 235, 124276, 2021; Journal of Colloid and Interface Science 628, 261-270, 2022; Polymers 14 (13), 2521, 2022; Journal of Colloid and Interface Science 599, 443-452, 2021; Frontiers in Chemistry 8, 89, 2020; Journal of Colloid and Interface Science 609, 179-187,2022; etc.
11. There are still some typos and grammar issues in the manuscript. Author should carefully recheck the whole manuscript.
Reviewer 2 Report
This paper entitled ‘Highly porous carbon aerogels for high-performance supercapacitor electrodes’ has demonstrated carbon aerogel-based porous supercapacitors electrodes were fabricated by polymerization and physical activation processes. The fabricated ACA has good performance. I think this paper is well-written and well-structured. This topic is quite interesting to the community and the authors have also provided sufficient background and included most of the relevant references in the discussions. I recommend this manuscript to be accepted after addressing the following minor points as shown below.
1. Name of ‘ACA60’, ‘ACA75’, etc. should be explained at the first time when they are used.
2. In Line 117, ‘77 K/N2’ should be ‘N2/77 K’, the same as Line 113
3. What are the hollow circles in figure 3(a) representing for? Why are they increasing at different pressure for different samples?
4. What’s the x-axils and y axils for the inset in figure 3(b)? And the explanation for the inset in figure 3(b) is missing.
5. What factors contribute to the high specific capacitances of the ACA electrodes? What role do micro- or mesoporous poles play in the rate capability and cycling stability of the ACA electrodes?
6. What were the specific capacitance, rate capability, and cycling stability achieved by the ACA electrodes under 1-5 A g-1 after 3000 cycles?
Reviewer 3 Report
In this work, the authors have prepared porous CAs activated by gaseous carbon dioxide, which show a high specific surface areas of 2503 m2 g–1 and achieved a specific gravimetric capacitance of up to 89.1 F g–1 at a current density of 1 A g–1, along with a high capacitance retention of 93.2% after 3000 cycles. The result and discussion are reasonable and acceptable. However, there are still some places which should be carefully issued.
1. TEM images should be added to character the pores of the Cas.
2. In this paper, only three-electrode system was used to investigate the electrochemical performances. Two electrode coin-cell test should be added to calculate the performance of supercapacitors.
3. EIS was missed in this manuscript.
4. The electrochemical performances should be compared with other reported activated carbon aerogels.
Reviewer 4 Report
In this work, the author reported “Highly porous carbon aerogels for high-performance supercapacitor electrodes" and systematically characterized using various physiochemical techniques. I strongly believe that the research part has been experimented with in a proper procedure. Therefore, I recommended this work for publication in the nanomaterial journal. However, some of the major concerns should be addressed before proceeding with further actions.
1. Abstract: Authors should include some results obtained from the characterization data (SEM morphology)
2. Abstract requires more technical achievements from the proposed work to highlight the novelty of the work.
3. Authors should discuss the contribution of Oxygen functionalities in the capacitive performance of the CA samples. Because, the obtained CV is not a perfect rectangle.
4. What is the mass loading of active materials on the working electrode?
5. The obtained specific capacitance is comparatively low than the many reported Carbon aerogel materials. Authors should justify.
6. Fig.4, the x-axis title should be “Potential vs. Ag/AgCl (V)”
7. Authors should include EIS analysis to evaluate the resistance of the fabricated electrode.
8. The authors should include RAMAN, XPS analysis to confirm the formation the CAs.
Reviewer 5 Report
The author demonstrated a 3D carbon aerogel as an electrode with a high surface area and large pore volume for supercapacitors. The manuscript is well-written and the experimental part is comprehensive. I recommend the major revision before acceptance.
1. What are the mechanical properties of the aerogel electrode?
2. Please give more details about the electrolyte used.
3. What is the mass loading of the material in the device?
4. Please compare the electrochemical results with the published data.
Round 2
Reviewer 1 Report
Accept in present form
Reviewer 3 Report
All my questions have been issued. I suggest accepted.
Reviewer 4 Report
The authors have made the necessary improvements in the revised file. Now this revised manuscript can be published in its present form.